# Physical Activity and Sedentary Lifestyle in University Students: Changes during Confinement Due to the COVID-19 Pandemic

**DOI:** 10.3390/ijerph17186567

**Published:** 2020-09-09

**Authors:** Cristina Romero-Blanco, Julián Rodríguez-Almagro, María Dolores Onieva-Zafra, María Laura Parra-Fernández, María del Carmen Prado-Laguna, Antonio Hernández-Martínez

**Affiliations:** Department of Nursing, Physiotherapy and Occupational Therapy, Ciudad Real Faculty of Nursing, University of Castilla-La Mancha, 13071 Ciudad Real, Spain; Cristina.Romero@uclm.es (C.R.-B.); MariaDolores.Onieva@uclm.es (M.D.O.-Z.); MariaLaura.Parra@uclm.es (M.L.P.-F.); Carmina.Prado@uclm.es (M.d.C.P.-L.); Antonio.HMartinez@uclm.es (A.H.-M.)

**Keywords:** college student, physical activity, sedentary behaviour, COVID-19, lockdown, healthy lifestyle

## Abstract

Regular physical activity is related to many factors in a university student’s environment. The coronavirus pandemic and the resulting lockdown have restricted many elements of our environment. The aim of this study was to evaluate students’ physical activity and sedentary behaviour at two points in time: before and during the coronavirus lockdown. As a secondary aim, we also wanted to look at changes resulting from other factors (alcohol, tobacco, diet, stages of change, symptoms of anxiety/depression and sociodemographic characteristics). We conducted an observational, cross-sectional, pre-post study with two cut-off points. Two hundred and thirteen students took part in the study. The main dependent variables were physical activity and sitting time, measured using the International Physical Activity Questionnaire—Short Form (IPAQ-SF). Parametric and non-parametric tests were used for paired and unpaired data, as well as group-stratified analysis. During lockdown, both weekly physical activity (MD: −159.87; CI: −100.44, −219.31) and weekly sitting time increased (MD: −106.76; CI: −71.85, −141.67). In the group analysis, differences were observed in relation to gender, year of study, BMI, alcohol consumption, tobacco use, symptoms of anxiety/depression, Mediterranean diet, living situation and stage of change. The results showed an increase in both physical activity and sitting time globally and by group.

## 1. Introduction

A healthy lifestyle should be promoted among all ages, but the earlier a habit is formed, the more likely it is to become rooted [1]. Regular physical activity is one of the most effective ways of preventing premature death [2,3]. The World Health Organization (WHO) recommends at least 150 min of moderate physical activity, 75 min of vigorous activity, or a combination of the two, per week [4]. Independently of the physical activity carried out, it is important to assess sedentary behaviour (SB) as this is related to increased morbidity and cardiovascular risk factors [5]. By 2030, the WHO aims to reduce the prevalence of physical inactivity by 15% worldwide [6].

In Spain, the amount of physical activity carried out by university students is low [7] and is in many cases linked to other healthy habits such as eating fruit and not smoking [8]. Meanwhile, sedentary behaviour is a health problem in the child and youth population, which is aggravated with age [9]. In university students, sitting time can exceed 9 h a day [10].

It is known that individual factors such as age, sex and health status affect the physical activity that individuals do [11]. Other factors associated with physical activity are motivation, lack of time and aspects related to body image or physical appearance [12]; some of the beneficial effects of physical activity are reduced anxiety and depression [13,14]. However, there are several factors that come into play throughout an individual’s lifetime that can either facilitate or impede a behaviour, with the transition from secondary education to university being a decisive moment [15]. It is at this time that young adults form their behavioural habits, so the role of Healthy Universities and the healthy habits they acquire at this stage are fundamental in maintaining this behaviour in the years to come [16].

When it comes to making physical activity a regular habit, the elements that may be related have been studied in depth [17]. Ecological models are considered one of the most significant theoretical approaches when it comes to analysing habit formation [18]. These models establish that in addition to individual factors, social and environmental factors are determinant in forming and maintaining physical activity habits [19].

The COVID-19 pandemic led to the population being confined to their homes [20]. In Spain, from March to April 2020, there was a prohibition on going outside to engage in sporting or social activities. During this period, elements of the built environment and other factors related to individuals’ environments were restricted due to the state of alarm. This created a valuable opportunity to assess physical activity without taking these factors into account. Experts’ recommendations to prevent sedentary behaviour during lockdown included taking active breaks, getting up and walking around the house, and doing online workouts [21]. However, during the pandemic, an overall negative effect on physical activity intensity was observed, as well as a rise in the consumption of less healthy food and a 28.6% increase in sedentary behaviour [22]. A reduction in physical activity was also observed in university students [23], along with increased levels of anxiety among 18- to 34-year-olds [24].

Spanish university students had to continue attending classes online, and their social lives were limited due to the prohibition on going outside. During lockdown, physical activity could have been an opportunity to pass the time, or, conversely, sedentary behaviour could have increased. The other characteristics of each individual (gender, motivation, eating habits, mental state etc.) could have either facilitated or interfered with the decision to exercise. The hypothesis put forward was that students’ sedentary behaviour would have increased during lockdown since they were confined to their homes, and that their physical activity would have decreased since they could not go outside to exercise.

In this study, we aimed to analyse the physical activity university students did before and during lockdown. To broaden our approach, as a secondary aim, we also wanted to look at changes in physical activity and sedentary behaviour resulting from other factors such as alcohol and tobacco consumption, adherence to a Mediterranean diet, motivation, symptoms of anxiety/depression and sociodemographic characteristics. We aimed to evaluate whether there were any differences when certain factors affecting individuals’ environments were restricted.

## 2. Material and Methods

### 2.1. Design and Selection of Study Subjects

This was an observational, cross-sectional, pre-post study on health sciences students, with two cut-off points. The first cut-off point was between 15 and 30 January 2020, prior to the state of alarm being put in place, and the second sample point was between 1 and 15 April 2020. This study received the approval of the Ethics and Clinical Research Committee of Ciudad Real, in Spain, with protocol number (C-291, 11/2019).

This study was carried out within the context of another study that we conducted on healthy habits and lifestyles, with an estimated follow-up period of 9 months. Due to the state of alarm and lockdown, recruitment of subjects was temporarily suspended and a decision was made to study the impact of lockdown on the population already participating. There were no exclusion criteria, other than failure to fully complete the questionnaire.

To estimate the sample considering a bilateral hypothesis, the following criteria were used: variance in the pre-lockdown control group of 33,929.60, obtained using the total minutes of physical activity [25], a beta risk of 20% (power = 80%), a confidence level of 95% and a clinically important difference of 60 min with respect to the control group. It was therefore estimated that a minimum of 148 study subjects would be needed. Considering a missing values ratio of 20%, the resulting sample size would be 185 subjects.

The students invited to take part were first- to fourth-year students who agreed to respond to the questionnaire at both time points. The questionnaires were administered during the second university semester. The first data collection point was two weeks after the end of the exam period, while the second data collection point was four weeks into lockdown. At the second data collection point, students could not leave their homes except for essential purposes such as buying food or going to hospital. Outdoor exercise was prohibited across Spain; anyone breaching the rules faced a 600 euro fine. During lockdown, university classes continued online with the same schedule as usual. The university provided internet access or technological devices to any students who requested them so that they could continue attending classes. Online classes did not contain any recommendations for students to carry out physical activity.

### 2.2. Instruments and Procedures

An ad hoc self-administered questionnaire was used, collecting sociodemographic information such as sex, age, weight, height, place of residence during the academic year, smoking habits (yes/no and number of cigarettes per day) and alcohol consumption (yes/no and number of drinks per week). For perceived health status and the existence of problems with anxiety/depression, the EuroQol 5D (EQ-5D) questionnaire was used [26].

To assess adherence to the Mediterranean diet, the PREDIMED questionnaire [27] was used, which uses 14 questions to assess the frequency of food consumption and eating habits. Each question has a possible score of 0 or 1. The result allows classification into low adherence or high adherence.

Stages of change (SOC) in physical activity were assessed using Prochaska and DiClemente’s Transtheoretical Model (TTM) [28]. Five stages of motivation for change were evaluated: pre-contemplation (I don’t exercise and I don’t intend to), contemplation (I don’t exercise, but I’d like to), preparation (I exercise sometimes), action (I have been regularly exercising for less than 6 months) and maintenance (I have been regularly exercising for more than 6 months).

Physical activity was measured using the International Physical Activity Questionnaire—Short Form (IPAQ-SF), which contains 7 questions [29]. The questionnaire was used to obtain the total minutes of physical activity per week and sitting time per day.

### 2.3. Statistical Analysis

First, descriptive statistical analysis was performed using absolute and relative frequencies for categorical variables and mean with standard deviation (SD) for the quantitative variables. Next, bivariate analysis was performed on the whole sample for paired data between weekly minutes of physical activity for the two sample points (pre-lockdown and lockdown). We used the Kolmogorov–Smirnov test to verify the normality of the quantitative variables. Since there were variables that were not normally distributed, we then used the non-parametric Wilcoxon signed-rank test. We also used the parametric Student-Fisher *t*-test to evaluate whether there were statistical differences in some comparisons and to obtain an approximation of the differences found.

Finally, the same analyses were performed again, but this time stratified for different sub-groups. Mean differences (MD) were obtained with a confidence interval of 95% (CI). All calculations were done using the program SPSS v24.0 (IBM Corp, New York, NY, USA).

## 3. Results

Two hundred and thirteen health sciences students participated in this study. The mean age was 20.5 years (SD = 4.56). Of the participants, 80.8% (172) were women, 76.5% (163) were normal weight and 9.9% (21) were smokers. The rest of the demographic characteristics and health parameters are shown in Table 1.

Then, the results of the IPAQ questionnaire were analysed: days and minutes of physical activity per week, as well as time spent sitting per week at both time points studied (Table 2). We observed a significant increase in the number of days on which students engaged in physical activity, both vigorous (mean difference (MD) −1.21; 95%CI −0.94, −1.49) and moderate (MD −1.41; 95%CI −1.13, −1.70), as well as the total number of minutes of physical activity per week (MD −159.87; 95%CI −100.44, −219.31). During lockdown, daily sitting time also increased by 141.67 min (95%CI: −71.85, −141.67).

We then analysed physical activity by group (Table 3). When we looked at the differences in average minutes of physical activity, all groups analysed spent more time doing physical activity during lockdown (although not all of them significantly). Groups that showed significant differences were women; first, second and third year of study; normal or low BMI; and those who did not eat a Mediterranean diet. Average physical activity time reduced during lockdown for participants in the pre-contemplation (MD: 37.50; 95% CI: −115.33, 190.33) and contemplation (MD: 31.08; 95%CI: −15.87, 78.03) stages. In other words, they spent less time on physical activity, although this difference was not significant. Conversely, for those in the preparation (MD: −75.59; 95%CI: −0.92, −150.25) and action (MD: 322.69; 95%CI: −214.84, −430.55) stages, significant differences (*p* < 0.05) were observed. In the rest of the groups analysed, statistically significant differences were observed between the two time points, except for men, final-year students, those that were overweight or obese and those that ate a Mediterranean diet.

Finally, the analysis by group (Table 4) showed significant differences (*p* < 0.05) in sitting time before and during lockdown in all groups except first-year students, those that were overweight or obese, smokers and those in the pre-contemplation stage. Sitting time increased in all groups of the variables gender, alcohol, symptoms of anxiety/depression and Mediterranean diet. It also increased in the following groups: second, third and fourth year of study; normal and underweight BMI; non-smokers; those living in a university residence, shared apartment or with family; and those in the contemplation, preparation, action and maintenance stages.

## 4. Discussion

This study aimed to evaluate physical activity and sedentary behaviour in health sciences students before and during the lockdown. At the first time point, students were in their normal study environment, while at the second, their social and environmental setting was limited due to lockdown. The results showed changes in physical activity and sedentary behaviour patterns both globally and by group. Overall, students spent more time doing physical activity and spent more time sitting when their usual environment was limited.

In the analysis by group, minutes of physical activity increased significantly during lockdown among the following groups: women; all years of study except final year; normal or low BMI; those who did not eat a Mediterranean diet; and those in the preparation or action stage of change. Sitting time increased in all groups of the variables gender, alcohol, symptoms of anxiety/depression and Mediterranean diet. The groups that did not experience differences were: first year of study, overweight or obese, smokers and those in the pre-contemplation stage. These four groups spent the most time sitting at the first data collection point when compared with the rest of their cohort; in other words, sedentary behaviour was already high before lockdown and there were no significant differences at the second data collection point.

Some researchers believed that lockdown would cause inactivity and an increase in sedentary behaviour and that measures would need to be taken to prevent these effects [30]. In fact, during lockdown, people modified their lifestyles, with an increase in sitting time due to people spending more time at home, and there was also a reduction in the amount of time spent on physical activity [22]. In our study, the initial hypothesis was partially confirmed: there was an increase in sitting time, but unexpectedly, there was also an increase in both the amount of time spent doing physical activity and the number of days on which participants were active. We expected to find an increase in sitting time due to the restrictions on movement; however, we also thought that the increase in screen time would reduce physical activity time, since in previous studies conducted in the Spanish university population, more screen time was associated with higher inactivity levels [31]. We do not know the exact reasons why physical activity increased, and we do not know if the effects on physical activity habits would have been maintained if the lockdown had gone on for longer. The environment in which students live affects their sedentary behaviour patterns [32], and it seems that the characteristics of health sciences students’ environments do not facilitate physical activity. Rather than being an obstacle, restricted social relations and not having access to the built environment in their community increased the number of days and minutes students spent doing physical activity. In the case of health sciences students, another factor to consider is that their training in promoting healthy habits may have influenced their decision to exercise at home.

No changes in physical activity were found in men. Perhaps men and women had different motivations and the environment influences one gender more strongly. In previous studies on motives for physical activity by gender [33], some variables that motivated men but not women were elements related to the environment, such as competition or social recognition, while weight control was the main motivation for women. In our study, women accounted for more than 80% of the sample, so the lack of results may also be due to the fact that there were fewer male participants.

The effect of the built environment is yet to be determined for those with a high BMI [34]. The data in this study show that in overweight or obese students, there were no changes in time spent doing physical activity or sitting time.

As we have seen, healthy habits that are ingrained in the population are not affected by the lockdown: this is the case of the Mediterranean diet [35]. In this study, we observed that students that ate a Mediterranean diet spent more time doing physical activity and that their physical activity patterns did not change significantly. This suggests that those that lead a healthy lifestyle pay attention to both diet and exercise and persist with their habits regardless of the environment. Conversely, those with unhealthy habits stick to them and experience no changes during lockdown. This is the case for smoking and sedentary behaviour. Grouping of healthy and non-healthy factors is habitual in university students [8,25]: those that are more sedentary are also more likely to smoke or spend a lot of time watching screens, while those that exercise regularly tend to eat more fruit and vegetables and drink less alcohol. Contrary to what we expected, smokers did spend more time doing physical activity during lockdown. It would be interesting to investigate the reasons for this. In our sample of the population, the percentage of smokers was very low, and the number of cigarettes smoked per day was also low, so we believe more research is needed in a sample with more smokers.

In our results, we also found differences based on year of study. Among final-year students, physical activity did not vary significantly. This group also spent the least time doing physical activity at both time points analysed. In their meta-analysis, Keating et al. indicate that with regard to year of study, the majority of studies find no differences in physical activity, but that some studies suggest that higher years of study are less active [36]. As for sedentary behaviour, it was observed that first-year students spent more time sitting and that lockdown did not bring about any significant changes. Some studies, contrary to the findings of our study, observed that students in higher years of study were more sedentary due to a higher workload [10]. In health sciences students, most of the theoretical workload is in the first year, while in their final year students spend most of their time on placement. Another possible factor could be that first-year students might have practiced sport in secondary school and kept up the habit. It would have been interesting to ask students about their sports histories.

In this study, we evaluated stages of change, one of the central concepts of the transtheoretical model of change. This model was initially used to treat tobacco and alcohol problems, but it was later adapted to other aspects of health such as physical activity and sedentary behaviour [37,38,39]. The analysis of the stages of change and how they affected the participants was very interesting. Participants in the first two stages did not experience any changes, and neither did those in the last stage. The behaviour of participants that exercised as part of their routine remained practically the same, as did the behavior of those that did not do any exercise. However, for those that were motivated but had not yet made exercise a regular habit, lockdown was a good opportunity to increase their dedication. In line with these findings, Di Renzo et al. [35] observed in a recent study that lockdown increased activity among people that did sport occasionally because they had more time at home, but those that did not do any exercise did not use the situation as an opportunity to start.

Overall, the results show that minutes of physical activity increased, as did minutes of sitting time. Although the results during lockdown are positive in terms of physical activity, it is necessary to recognise that this population might suffer from health issues in the future due to an increase in sedentary behaviour. It would be interesting to find out what the reasons were for students having this behaviour. Perhaps they realised that their sitting time increased (they were not walking to class, walking to their car, going shopping, standing up, going to their jobs etc.) and compensated for this with some high-intensity exercise. Another aspect that could have affected the results is that the students were involved in the health sciences field, so they may have been more prone to exercising during the pandemic than students in other majors such as engineering or literature. This is why we cannot exclusively consider the limitation of the environment during lockdown to be the cause of the changes in physical activity and sedentary behaviour. It would be interesting to continue studying the elements related to university students’ physical activity/sedentary behaviour and their surroundings in order to plan strategies that promote an increase in physical activity levels in this group.

### Strengths and Limitations

Our study has various limitations that should be considered. Firstly, it is an observational study and all study subjects volunteered to participate in the questionnaire, so there may be a selection bias. Secondly, we did not measure whether there was any risk of exposure to COVID-19 infection, a factor that could have influenced our assessment of physical activity and sedentary behaviour.

Another limitation is the use of a self-administered questionnaire to evaluate physical activity and sedentary behaviour. It would have been more interesting to perform a real assessment of physical activity using accelerometry and also investigate their sports history. This could be a future line of research.

Finally, the lack of significance in some of the strata analysed could be due to a lack of statistical power because of the low number of subjects in some groups. Furthermore, we do not know if these changes in physical activity would have been maintained if lockdown had gone on longer.

As for the strengths, this is the first study to look at physical activity and sedentary behaviour in university students studying health sciences both before and during lockdown.

## 5. Conclusions

In this study, we observed the behaviour of health sciences students when deprived of their usual social and community environment. Participants spent more time doing physical activity and also spent more time sitting. University students’ social environment may be a barrier to building an exercise habit, especially among women, and motivation seems to have a significant bearing on whether university students engage in physical activity. More efforts should be made to create strategies that motivate students to lead a healthy lifestyle in all aspects (diet, avoiding harmful substances, mental health etc.), with a particular emphasis on engaging in physical activity and reducing sitting time.

## Figures and Tables

**Table 1 ijerph-17-06567-t001:** Descriptive characteristics of the participants.

	% (n)	Mean (SD)
**Age**		20.5 (4.56)
**Gender**		
Male	19.2 (41)	
Female	80.8 (172)	
**Year of study**		
First year	39.0 (83)	
Second year	29.6 (63)	
Third year	21.1 (45)	
Fourth year	10.3 (22)	
**BMI**		22.11 (3.54)
Normal	76.5 (163)	
Underweight	12.2 (26)	
Overweight or Obese	11.3 (24)	
**Alcohol Consumption**		2.25 (1.66) ^a^
Yes	81.7 (174)	
No	18.3 (39)	
**Smoker**		0.67 (2.46) ^b^
Yes	9.9 (21)	
No	90.3 (192)	
**Health status (scale 0–100)**		78.76 (14.96)
**Anxiety/Depression**		
Yes	24.9 (53)	
No	75.1 (160)	
**Mediterranean Diet**		7.65 (1.79) ^c^
Yes	31.0 (66)	
No	69.0 (147)	
**Living situation**		
University residence	18.8 (40)	
Shared apartment	42.7 (91)	
With family	33.4 (71)	
Other	5.1 (11)	
**Stage of change**		
Pre-contemplation	1.9 (4)	
Contemplation	11.3 (24)	
Preparation	23.9 (51)	
Action	41.3 (88)	
Maintenance	21.6 (46)	

SD: Standard deviation; ^a^: number of alcoholic drinks per week; ^b^: cigarettes per day; ^c^: PREDIMED score.

**Table 2 ijerph-17-06567-t002:** International Physical Activity Questionnaire (IPAQ) scores before and during lockdown.

	PRE		POST						
	Mean	SD	Mean	SD	MD	CI 95%		*p*-Value Wilcoxon	*p*-Value *t*-Student
Days of vigorous PA	0.98	1.33	2.19	2.02	−1.21	−1.49	−0.94	**<0.001**	**<0.001**
Days of moderate PA	1.74	1.56	3.15	2.05	−1.41	−1.70	−1.13	**<0.001**	**<0.001**
Minutes of vigorous PA	28.47	54.13	30.66	30.94	−2.68	−9.71	4.35	0.007	0.45
Minutes of moderate PA	42.81	48.44	47.74	50.80	−4.93	−12.36	2.49	0.353	0.19
Total minutes of weekly PA	223.30	305.47	383.17	438.90	−159.87	−219.31	−100.44	**<0.001**	**<0.001**
Daily sitting time	418.59	201.58	525.35	194.57	−106.76	−141.67	−71.85	**<0.001**	**<0.001**

PA: physical activity; Bold: *p* < 0.05.

**Table 3 ijerph-17-06567-t003:** Physical activity: total minutes per week pre and post by group.

	Pre	Post	MD	95% CI		*p*	*p* Wilcoxon
	(Mean ± SD)	(Mean ± SD)				*t*-Student	
**Gender**							
Male	226.46 ± 250.10	279.93 ± 446.91	−53.46	−165.01	58.09	0.339	0.362
Female	222.55 ± 317.89	407.78 ± 404.76	−185.24	−253.80	−116.68	**<0.001**	**0.000**
**Year of study**							
First year	207.83 ± 290.89	463.36 ± 567.77	−255.53	−381.71	−129.35	**<0.001**	**<0.001**
Second year	234.84 ± 355.42	331.98 ± 301.27	−97.14	−166.41	−27.88	**0.007**	**0.002**
Third year	265.56 ± 292.99	409.82 ± 388.14	−144.27	−260.53	−28.00	**0.016**	**0.014**
Fourth year	162.18 ± 222.00	172.73 ± 145.18	−10.55	−101.43	80.34	0.812	0.590
**BMI**							
Normal	234.14 ± 313.41	400.62 ± 454,218	−166.48	−238.88	−94.07	**<0.001**	**<0.001**
Underweight	149.54 ± 234.27	396.92 ± 426.77	−247.38	−395.35	−99.42	**0.002**	**0.001**
Overweight or Obese	229.58 ± 318,686	249.79 ± 321.66	−20.21	−129.00	88.58	0.704	0.779
**Alcohol**							
Yes	240.22 ± 326.83	401.60 ± 431.19	−161.38	−228.60	−94.16	**<0.001**	**0.004**
No	147.82 ± 164.68	300.97 ± 468.77	−153.15	−283.45	−22.86	**0.022**	**0.000**
**Smoker**							
Yes	253.57 ± 356.09	524.71 ± 435.95	−271.14	−495.81	−46.48	**0.020**	**0.008**
No	219.05 ± 302.78	362.87 ± 437.43	−143.81	−207.93	−79.69	**<0.001**	**<0.001**
**Anxiety/Depression**							
Yes	269.96 ± 382.95	438.92 ± 522.22	−168.96	−329.86	−8.06	**0.040**	0.117
No	209.53 ± 275.98	366.20 ± 409.69	−156.67	−217.35	−95.99	**<0.001**	**<0.001**
**Mediterranean Diet**							
Yes	352.55 ± 391,454	427.95 ± 362,224	−75.41	−170.61	19.80	0.119	0.090
No	165.27 ± 237.35	363.07 ± 469.02	−197.80	−272.45	−123.14	**<0.001**	**<0.001**
**Living situation**							
University residence	197.25 ± 233,384	515.00 ± 498.81	−317.75	−482.22	−153.28	**<0.001**	**<0.001**
Shared apartment	204.87 ± 318.48	331.97 ± 363.60	−127.10	−206.03	−48.17	**0.002**	**<0.001**
With family	267.40 ± 327.12	383.99 ± 485.21	−116.58	−220.80	−12.36	**0.029**	**0.028**
Other	88.00 ± 155.95	248.00 ± 281.53	−160.00	−620.15	300.15	0.389	0.225
**Stage of Change**							
Pre-contemplation	45.00 ± 90.00	7.50 ± 15.00	37.50	−115.33	190.33	0.492	0.655
Contemplation	69.58 ± 128.82	38.50 ± 87.04	31.08	−15.87	78.03	0.184	0.386
Preparation	126.86 ± 194.61	202.45 ± 269.94	−75.59	−150.25	−0.92	**0.047**	**0.007**
Action	199.86 ± 315.70	522.56 ± 490.71	−322.69	−430.55	−214.84	**<0.001**	**<0.001**
Maintenance	470.76 ± 325.86	529.39 ± 429.34	−58.63	−201.93	84.66	0.414	0.502

MD: Mean Difference; SD: Standard Deviation; Bold: Statistically significant differences.

**Table 4 ijerph-17-06567-t004:** Sitting time: daily minutes pre and post by group.

	Pre	Post	MD	95% CI		*p*-Value *t*-Student	*p*-Value Wilcoxon
	(Mean ± SD)	(Mean ± SD)					
**Gender**							
Male	403.90 ± 194.90	545.85 ± 230.65	−141.95	−214.93	−68.97	**<0.001**	**0.001**
Female	422.09 ± 203.54	520.47 ± 185.37	−98.37	−138.18	−58.57	**<0.001**	**<0.001**
**Year of study**							
First year	465.54 ± 236.42	520.84 ± 191.10	−55.30	−119.07	8.47	0.088	0.062
Second year	449.52 ± 148.49	526.67 ± 198.32	−77.14	−127.97	−26.32	**0.004**	**0.005**
Third year	373.33 ± 181.76	520.67 ± 215.50	−147.33	−220.70	−73.96	**<0.001**	**0.001**
Fourth year	245.45 ± 101.26	548.18 ± 159.60	−302.73	−390.57	−214.89	**<0.001**	**<0.001**
**BMI**							
Normal	419.63 ± 209.39	525.09 ± 194.80	−105.46	−145.85	−65.07	**<0.001**	**<0.001**
Underweight	385.38 ± 186.81	510.00 ± 187.64	−124.62	−221.29	−27.94	**0.014**	**0.009**
Overweight or Obese	447.50 ± 160.20	543.75 ± 206.77	−96.25	−209.98	17.48	0.093	0.109
**Alcohol**							
Yes	412.07 ± 209.04	521.03 ± 191.46	−108.97	−148.12	−69.81	**<0.001**	**<0.001**
No	447.69 ± 163.36	544.62 ± 209.38	−96.92	−176.82	−17.03	**0.019**	**0.029**
**Smoker**							
Yes	454.29 ± 141.44	460.00 ± 202.21	−5.71	−93.01	81.58	0.893	0.968
No	412.75 ± 179.80	538.02 ± 194.16	−125.27	−160.79	−89.76	**<0.001**	**<0.001**
**Anxiety/Depression**							
Yes	441.92 ± 274.58	526.15 ± 173.56	−84.23	−170.30	1.84	0.055	**0.004**
No	413.63 ± 169.38	525.00 ± 202.02	−111.38	−148.66	−74.09	**<0.001**	**<0.001**
**Mediterranean Diet**							
Yes	406.36 ± 181.01	534.09 ± 190.12	−127.73	−187.13	−68.33	**<0.001**	**<0.001**
No	424.08 ± 210.52	521.43 ± 197.05	−97.35	−140.69	−54.00	**<0.001**	**<0.001**
**Living situation**							
University residence	408.00 ± 163.73	510.75 ± 181.72	−102.75	−174.74	−30.76	**0.006**	**0.015**
Shared apartment	401.54 ± 183.67	502.42 ± 191.97	−100.88	−152.23	−49.53	**<0.001**	**<0.001**
With family	448.05 ± 232.09	568.44 ± 201.18	−120.39	−185.47	−55.31	**<0.001**	**<0.001**
Other	360.00 ± 284.60	396.00 ± 124.42	−36.00	−439.26	367.26	0.816	0.786
**Stage of Change**							
Pre-contemplation	510.00 ± 311.77	660.00 ± 272.76	−150.00	−332.82	32.82	0.080	0.102
Contemplation	422.50 ± 163.58	600.00 ± 212.31	−177.50	−271.55	−83.45	**0.001**	**0.001**
Preparation	403.53 ± 190.15	538.82 ± 180.69	−135.29	−205.86	−64.73	**0.000**	**0.001**
Action	414.55 ± 182.32	498.07 ± 189.98	−83.52	−136.18	−30.86	**0.002**	**0.007**
Maintenance	433.04 ± 256.10	511.96 ± 194.57	−78.91	−169.72	11.89	0.087	**0.006**

MD: Mean Difference; SD: Standard Deviation; Bold: Statistically significant differences.

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
