# Peer review of "Physical Activity and Sedentary Lifestyle in University Students: Changes during Confinement Due to the COVID-19 Pandemic"

_ijerph, 2020, doi:10.3390/ijerph17186567_

Round 1

Reviewer 1 Report

Comments, Suggestions and Questions for Authors

Abstract

In addition to the main objective “The aim of this study was to evaluate students’ physical activity and sedentary behaviour at two points in time: before and during the coronavirus lockdown”, it would be pertinent to insert the secondary objective in relation to group analyzes.

Line 33-34: “The results show the association between lockdown…”. Correlation/association analyzes were not performed. The purpose of the study is not to verify the association of variables, right?

The keywords you added are correct. However, some of them repeat words from the title, which is not advisable.

Introduction

There are many paragraphs in the introduction. I advise a better structuring of it.

Line 48: “Other factors associated with physical activity are motivation, lack of time, or aspects related to body image or physical appearance; some of the beneficial effects of physical exercise are reduced anxiety and depression”. There seems to be confusion between the terms “physical activity” and “physical exercise”. Remember that they are different concepts, we should not talk about physical activity and, in the same sentence, physical exercise.

The introduction addresses the importance of physical activity, the influence of the surrounding environment, and the lockdown caused by COVID-19. However, scientific evidence related to sedentary behavior is not addressed. It is important to clarify the difference between sedentary behavior and physical inactivity. Since the main objective of the study is to evaluate students’ physical activity and sedentary behaviour at two points in time (before and during the coronavirus lockdown), it is relevant to address sedentary behavior in the introduction.

One more time: it is also important to indicate, in a more developed way, the value or potential meaning of the problem area. Also, it would be interesting for the reader to present and substantiate the hypotheses.

Material and Methods

I suggest adding a subtitle for the instruments and procedures.

Results

How were the Mean and standard deviation of the variable "alcohol" and "tobacco" calculated, being qualitative variables?

Table 1: in the title, I suggest to change replace “population” to “participants”.

Depending on the normality of the data, the authors must present the results of the t-test (for normal data) or Wilcoxon (for non-normal data), not both simultaneously.

Discussion

The discussion needs to be more stringent and connected with the other sections. In addition, there are several other limitations of the study, such as lack of control over other variables. Again, it would be pertinent to add suggestions for future studies.

Overall: the authors collected multiple sets of data which makes the study confusing. Therefore, I suggest a substantial improvement in the introduction and discussion, in order to consider the article for acceptance.

Author Response

Dear editors:

Thank you for reviewing our manuscript titled “Limiting the environment in university students modifies physical activity and sedentary behaviour patterns”.

In the attached report, a complete account of all the points raised by the reviewer is given. All the recommendations have been followed and the necessary clarifications are made in the same order.

In case there is need for further clarification, we are fully willing to meet your demands.

Finally, we would like to thank the reviewers for their diligent work.

Looking forward to hearing from you.

Kind regards,

REVIEWER. #1

Abstract

In addition to the main objective “The aim of this study was to evaluate students’ physical activity and sedentary behaviour at two points in time: before and during the coronavirus lockdown”, it would be pertinent to insert the secondary objective in relation to group analyzes.

Response: The secondary objective has been added in line with your recommendation.

Line 33-34: “The results show the association between lockdown…”. Correlation/association analyzes were not performed. The purpose of the study is not to verify the association of variables, right?

Response: Thank you for your observation; that’s right, correlations were not studied. The sentence has been modified.

The keywords you added are correct. However, some of them repeat words from the title, which is not advisable.

 Response: We have changed this section in line with your recommendation. Thank you very much.

Introduction

There are many paragraphs in the introduction. I advise a better structuring of it.

Response: The introduction has been restructured. If you still think it could be improved, please let us know.

Line 48: “Other factors associated with physical activity are motivation, lack of time, or aspects related to body image or physical appearance; some of the beneficial effects of physical exercise are reduced anxiety and depression”. There seems to be confusion between the terms “physical activity” and “physical exercise”. Remember that they are different concepts, we should not talk about physical activity and, in the same sentence, physical exercise.

Response: Thank you for your observation. We have modified this sentence.

The introduction addresses the importance of physical activity, the influence of the surrounding environment, and the lockdown caused by COVID-19. However, scientific evidence related to sedentary behavior is not addressed. It is important to clarify the difference between sedentary behavior and physical inactivity. Since the main objective of the study is to evaluate students’ physical activity and sedentary behaviour at two points in time (before and during the coronavirus lockdown), it is relevant to address sedentary behavior in the introduction.

One more time: it is also important to indicate, in a more developed way, the value or potential meaning of the problem area. Also, it would be interesting for the reader to present and substantiate the hypotheses.

Response: Following your recommendations, we have included the concepts of sedentary behaviour and physical activity independently in the introduction, and have provided data relating to our country (Spain).

Material and Methods

I suggest adding a subtitle for the instruments and procedures.

Response: We have included this subtitle in line with your suggestion.

Results

How were the Mean and standard deviation of the variable "alcohol" and "tobacco" calculated, being qualitative variables?

Response: We asked participants if they consumed alcohol or tobacco and then asked them how much they consumed. That is why the quantitative data was included. For the statistical analysis by group at each time point (before and during lockdown), the yes/no response was used. This has been specified in the methodology section.

Table 1: in the title, I suggest to change replace “population” to “participants”.

Response: This has been modified. Many thanks.

Depending on the normality of the data, the authors must present the results of the t-test (for normal data) or Wilcoxon (for non-normal data), not both simultaneously.

Response: The data obtained in the IPAQ questionnaire are not normally distributed, so we used non-parametric tests. However, the problem with non-parametric tests is that they do not allow information to be obtained on the difference between groups with confidence intervals, they only provide information on p-values. This is the reason why we included the Student-Fisher t-test for paired data. This way, not only can readers see that there are statistical differences in some comparisons, they  can also get an approximate idea of the difference found. As the use of both tests is quite unusual, this has been explained in the methodology section.

Discussion

The discussion needs to be more stringent and connected with the other sections. In addition, there are several other limitations of the study, such as lack of control over other variables. Again, it would be pertinent to add suggestions for future studies.

Overall: the authors collected multiple sets of data which makes the study confusing. Therefore, I suggest a substantial improvement in the introduction and discussion, in order to consider the article for acceptance.

Response: Thank you for your comments. We have substantially modified the introduction and discussion.

Reviewer 2 Report

This is an interesting study. Some questions;

The population is very selected, health science students and it would be of interest to know what the education during the lock down included. This might have encouraged the students to move.

Anxiety/depression should be changed to symptoms of anxiety/depression.

Minutes in MVPA increased as well as sedentary time. Most likely the students realised that their sitting time increased and compensated this with some high intensity exercise. PA at lower activity levels subsequently decreased. So the total time in acitivity decreased. This has probably nothing to do with the usual environment the students are in.The authors should add something about this in the discussion.

Author Response

Dear editors:

Thank you for reviewing our manuscript titled “Limiting the environment in university students modifies physical activity and sedentary behaviour patterns”.

In the attached report, a complete account of all the points raised by the reviewer is given. All the recommendations have been followed and the necessary clarifications are made in the same order.

In case there is need for further clarification, we are fully willing to meet your demands.

Finally, we would like to thank the reviewers for their diligent work.

Looking forward to hearing from you.

Kind regards,

REVIEWER. #2

This is an interesting study. Some questions;

The population is very selected, health science students and it would be of interest to know what the education during the lock down included. This might have encouraged the students to move.

Response: During this period, theoretical classes continued online. The content of the subjects would not have encouraged the students to move and the university did not make any recommendations in this regard. This has been explained in the methodology.

Anxiety/depression should be changed to symptoms of anxiety/depression.

Response: This has been modified in line with your suggestion.

Minutes in MVPA increased as well as sedentary time. Most likely the students realised that their sitting time increased and compensated this with some high intensity exercise. PA at lower activity levels subsequently decreased. So the total time in acitivity decreased. This has probably nothing to do with the usual environment the students are in.The authors should add something about this in the discussion.

Response: Thank you for your observation. We have added this to the discussion in line with your suggestion.

Reviewer 3 Report

Your manuscript addressed some of the important concerns that surround physical activity and sedentary behavior during a pandemic. There were some limitations that were not directly addressed, including your sample was made up of health science students. Due to these students already being involved in the health science field, they may be more prone to exercising during the pandemic than students in other majors such as engineering or literature. It would be important to address this in your manuscript, as future researchers may want to examine this topic with a diverse student population (students from different majors). 

You also mention an increase in both sedentary behavior and physical activity. In your discussion, it seems like there is some explanation that might be lacking. Though there are plenty of references throughout your discussion, there doesn't appear to be any in-depth examination of "why" these findings occurred. For example, this increase in both sedentary behavior and physical activity may be due to individuals exercising more but walking or standing less. They were not walking to class, walking to their car, going to stores, standing up, going to their jobs, etc. Therefore, they may have had more structured exercise which counted towards overall physical activity, but other than the minutes/hours they spent exercising each day, they might have been sitting the rest of the time due to quarantine.

Your results showed that there were no changes in men. I wonder if this could be due to the small sample size of specifically men in your study. This difference in sample size between men and women may be important to address. 

For seniors being more sedentary than freshman, I wonder if this is due to freshman being right out of high school where they may have been in a sport. Specifically because you mention that for the health sciences program freshman typically have a larger course load than seniors, I think that there could be a sports component to it or something related to freshman being only out of high school for 6 months. One limitation to your study might be sports history. Did you collect this information from the participants? It would be interesting to determine in any of these college students were also college athletes or played any sports. This would definitely be a factor to take into consideration when looking at physical activity levels pre and post quarantine. 

For the future, it would interesting to ask these individuals why they participated in more physical activity. What motivated them? ... Just a thought for future research so that we can understand the motivations behind exercising during a pandemic.

Finally, I simply have a couple of comments on the alcohol and smoking component. For smokers, you found that (contrary to existing literature) their physical activity levels increased. Could this be due to the smokers in your study not being chronic smokers? It appears from your table that for the 9.9% of smokers in your sample size, the average amount of cigarettes smoked per day was 0.67. This indicates that the average smoker did not even smoke 1 cigarette per day. This leads me to ask if these smokers were social smokers or did not smoke regularly. This would absolutely make a difference in how you would interpret your findings. In the future, it might be beneficial to ask if someone was a smoker and attach a number to it (e.g., "Do you smoke at least 1 cigarette per day?"). If you asked that question, the percentage of smokers in your sample would decrease; however, this is just a suggestion. 

For the alcohol component, I am not quite understanding what it means in the table where it reads "drink alcohol every time they go out." I am not sure what the average number of 2.25 means. Does that mean that these individuals drink an average of 2.25 drinks every time they go out? Are you determining if an individual drinks alcohol based on the number of drinks they drink or based on if they drink every single time they go out? I think this may be the wrong question to ask, as students can still drink alcohol without going out. Additionally, just because someone doesn't drink every time they go out doesn't mean they don't drink regularly at home. Maybe I am not understanding this correctly. I would like just a little more clarity on this. 

I hope that these comments are helpful and I enjoyed reading your manuscript. Thank you. 

Author Response

Dear editors:

Thank you for reviewing our manuscript titled “Limiting the environment in university students modifies physical activity and sedentary behaviour patterns”.

In the attached report, a complete account of all the points raised by the reviewer is given. All the recommendations have been followed and the necessary clarifications are made in the same order.

In case there is need for further clarification, we are fully willing to meet your demands.

Finally, we would like to thank the reviewers for their diligent work.

Looking forward to hearing from you.

Kind regards,

REVIEWER. #3

Your manuscript addressed some of the important concerns that surround physical activity and sedentary behavior during a pandemic. There were some limitations that were not directly addressed, including your sample was made up of health science students. Due to these students already being involved in the health science field, they may be more prone to exercising during the pandemic than students in other majors such as engineering or literature. It would be important to address this in your manuscript, as future researchers may want to examine this topic with a diverse student population (students from different majors). 

Response: Many thanks for your comment. We agree with you. We have added your suggestions to the discussion section.

You also mention an increase in both sedentary behavior and physical activity. In your discussion, it seems like there is some explanation that might be lacking. Though there are plenty of references throughout your discussion, there doesn't appear to be any in-depth examination of "why" these findings occurred. For example, this increase in both sedentary behavior and physical activity may be due to individuals exercising more but walking or standing less. They were not walking to class, walking to their car, going to stores, standing up, going to their jobs, etc. Therefore, they may have had more structured exercise which counted towards overall physical activity, but other than the minutes/hours they spent exercising each day, they might have been sitting the rest of the time due to quarantine.

Response: Thank you for your comments. We have used these to improve the discussion section.

Your results showed that there were no changes in men. I wonder if this could be due to the small sample size of specifically men in your study. This difference in sample size between men and women may be important to address. 

Response: Yes, you are right. There were less men than women and this could be the reason no changes were found in men. This has been mentioned in the discussion section.

For seniors being more sedentary than freshman, I wonder if this is due to freshman being right out of high school where they may have been in a sport. Specifically because you mention that for the health sciences program freshman typically have a larger course load than seniors, I think that there could be a sports component to it or something related to freshman being only out of high school for 6 months. One limitation to your study might be sports history. Did you collect this information from the participants? It would be interesting to determine in any of these college students were also college athletes or played any sports. This would definitely be a factor to take into consideration when looking at physical activity levels pre and post quarantine. 

Response: We agree with you and have included this in the discussion and study limitations.

For the future, it would interesting to ask these individuals why they participated in more physical activity. What motivated them? ... Just a thought for future research so that we can understand the motivations behind exercising during a pandemic.

Response: It would indeed be very interesting to find out what their motivations were. We intend to continue studying university students’ habits, so we will include this as a future research topic.

Finally, I simply have a couple of comments on the alcohol and smoking component. For smokers, you found that (contrary to existing literature) their physical activity levels increased. Could this be due to the smokers in your study not being chronic smokers? It appears from your table that for the 9.9% of smokers in your sample size, the average amount of cigarettes smoked per day was 0.67. This indicates that the average smoker did not even smoke 1 cigarette per day. This leads me to ask if these smokers were social smokers or did not smoke regularly. This would absolutely make a difference in how you would interpret your findings. In the future, it might be beneficial to ask if someone was a smoker and attach a number to it (e.g., "Do you smoke at least 1 cigarette per day?"). If you asked that question, the percentage of smokers in your sample would decrease; however, this is just a suggestion. 

Response: We agree with your suggestion. In this population sample, there were very few smokers and the number of cigarettes smoked was also low. This has been included in the discussion.

For the alcohol component, I am not quite understanding what it means in the table where it reads "drink alcohol every time they go out." I am not sure what the average number of 2.25 means. Does that mean that these individuals drink an average of 2.25 drinks every time they go out? Are you determining if an individual drinks alcohol based on the number of drinks they drink or based on if they drink every single time they go out? I think this may be the wrong question to ask, as students can still drink alcohol without going out. Additionally, just because someone doesn't drink every time they go out doesn't mean they don't drink regularly at home. Maybe I am not understanding this correctly. I would like just a little more clarity on this. 

Response: All of the students were asked if they drank alcohol. Then, they were asked how many drinks they consumed per week. We didn’t ask them to specify whether this was at home or when they went out. So, table I contains an error: we have changed “every time they go out” to “weekly”.

I hope that these comments are helpful and I enjoyed reading your manuscript. Thank you. 

Response: Many thanks for all your contributions. We are sure they will help us to improve the manuscript.

Reviewer 4 Report

This paper is well written and addresses an important public health problem (physical activity/sedentary behaviour), during this period of worldwide uncertainty we are all living, from which no one knows what will be the physical, mental, social and health consequences.

Broad comments: 

The authors should report the results from their study in the past tense; please change it throughout the entire manuscript.

I believe the discussion would improve if the authors address and acknowledge that, despite these results showing an increase in PA, it is necessary to recognize that this population might suffer from health issues due to an increase of the SB, in the future.

The conclusions should also acknowledge the need to on addressing gender inequalities, as showed in this study results. Please include this information.

Specific comments:

Page 3, line 112: EQ-5D questionnaire - Spell it out as this is the first time being used.

Page 4, line 140: were of normal weight – delete “of”.

Page 4, line 149: for the 3rd and 4th paragraphs of the results section, I would suggest the authors state the actual differences found rather than the exceptions. It is easier for the reader to capture real information in this way.

Page 4, line 153: average times reduced during lockdown – change for “average times of physical activity reduced during lockdown”.

Page 8, line 209: at two different points in time - suggest including “(before and during lockdown)”.

Page 9, line 214: in the 2nd and 3rd paragraphs of the discussion section, there is a lot of information that belongs to the results section, as it is written. I do not see the need to specify so much of the results in the discussion section, as this should actually be specified as it is in the results section. Therefore, I suggest rephrasing it into a shorter sentence.

Page 9, line 225: researchers had believed – delete “had”.

Page 9, line 225/226: cause inactivity and sedentary behaviour to increase – suggest rephrasing it into “cause inactivity and an increase in sedentary behaviour”.

Page 9, line 226: to prevent the effects of this – suggest rephrasing it into “to prevent these effects”.

Page 10, line 284/285: Finally, the lack of significance in some of the strata analysed may down to a lack of statistical power because of the low number of subjects in some groups. – this sentence is difficult to understand due to the use of the word “down”. Please rephrase it.

Page 10, line 288: university students studying health sciences - it seems more important to address the age range of the students than the topic they are studying.

Page 10, line 292: health sciences students – same as above; please focus on the age range of the sample.

P.S.: line numbers are according the PDF document.

Author Response

Dear editors:

Thank you for reviewing our manuscript titled “Limiting the environment in university students modifies physical activity and sedentary behaviour patterns”.

In the attached report, a complete account of all the points raised by the reviewer is given. All the recommendations have been followed and the necessary clarifications are made in the same order.

In case there is need for further clarification, we are fully willing to meet your demands.

Finally, we would like to thank the reviewers for their diligent work.

Looking forward to hearing from you.

Kind regards,

REVIEWER. #4

This paper is well written and addresses an important public health problem (physical activity/sedentary behaviour), during this period of worldwide uncertainty we are all living, from which no one knows what will be the physical, mental, social and health consequences.

Broad comments: 

The authors should report the results from their study in the past tense; please change it throughout the entire manuscript.

Response: Thank you for your observation, we have modified this.

I believe the discussion would improve if the authors address and acknowledge that, despite these results showing an increase in PA, it is necessary to recognize that this population might suffer from health issues due to an increase of the SB, in the future.

Response: This has been added to the discussion. Many thanks for your suggestion.

The conclusions should also acknowledge the need to on addressing gender inequalities, as showed in this study results. Please include this information.

Response: We have added this to the discussion in line with your instructions.

Specific comments:

Page 3, line 112: EQ-5D questionnaire - Spell it out as this is the first time being used.

Page 4, line 140: were of normal weight – delete “of”.

Page 4, line 149: for the 3rd and 4th paragraphs of the results section, I would suggest the authors state the actual differences found rather than the exceptions. It is easier for the reader to capture real information in this way.

Page 4, line 153: average times reduced during lockdown – change for “average times of physical activity reduced during lockdown”.

Page 8, line 209: at two different points in time - suggest including “(before and during lockdown)”.

Page 9, line 214: in the 2nd and 3rd paragraphs of the discussion section, there is a lot of information that belongs to the results section, as it is written. I do not see the need to specify so much of the results in the discussion section, as this should actually be specified as it is in the results section. Therefore, I suggest rephrasing it into a shorter sentence.

Page 9, line 225: researchers had believed – delete “had”.

Page 9, line 225/226: cause inactivity and sedentary behaviour to increase – suggest rephrasing it into “cause inactivity and an increase in sedentary behaviour”.

Page 9, line 226: to prevent the effects of this – suggest rephrasing it into “to prevent these effects”.

Page 10, line 284/285: Finally, the lack of significance in some of the strata analysed may down to a lack of statistical power because of the low number of subjects in some groups. – this sentence is difficult to understand due to the use of the word “down”. Please rephrase it.

Page 10, line 288: university students studying health sciences - it seems more important to address the age range of the students than the topic they are studying.

Page 10, line 292: health sciences students – same as above; please focus on the age range of the sample.

P.S.: line numbers are according the PDF document.

Response: We have modified the manuscript taking all of your suggestions into account. With regard to page 10, we agree with you, however, in the discussion we have explained that due to the nature of their studies, health sciences students may behave differently to students from other disciplines. It is for this reason that we thought it was important to highlight the topic they were studying.

Round 2

Reviewer 1 Report

I appreciate the authors’ efforts to revise the manuscript. While the paper has been greatly improved, I suggest that some changes need to be done before acceptance.

The abstract should be a total of about 200 words maximum.

One more time, it would be interesting for the reader to present and substantiate the hypotheses.

Some bibliographic references do not meet the standards.

Author Response

Dear editors:

Thank you for reviewing our manuscript titled “Limiting the environment in university students modifies physical activity and sedentary behaviour patterns”.

In the attached report, a complete account of all the points raised by the reviewer is given. All the recommendations have been followed and the necessary clarifications are made in the same order.

In case there is need for further clarification, we are fully willing to meet your demands.

Finally, we would like to thank the reviewers for their diligent work.

Looking forward to hearing from you.

Kind regards,

REVIEWER. #1

I appreciate the authors’ efforts to revise the manuscript. While the paper has been greatly improved, I suggest that some changes need to be done before acceptance.

The abstract should be a total of about 200 words maximum.

We appreciate the reviewer’s suggestion. The number of words in the abstract has been reduced.

One more time, it would be interesting for the reader to present and substantiate the hypotheses.

We agree with the reviewer’s corrections. The hypothesis has been included in the introduction and the results have been discussed in relation to the hypothesis in the discussion.

Some bibliographic references do not meet the standards.

The errors in the bibliographic references have been reviewed and corrected.
